# Whole genome sequencing of *Clostridioides difficile* PCR ribotype 046 suggests transmission between pigs and humans

**Anders Werner** [1]*, **Paula Mölling**[2], **Anna Fagerström**[2], **Fredrik Dyrkell**[3], **Dimitrios Arnellos**[3], **Karin Johansson**[2], **Martin Sundqvist**[2], **Torbjörn Norén**[2]

**1** Department of Clinical Microbiology, Sahlgrenska University Hospital, Göteborg, Region Västra Götaland, Sweden, **2** Faculty of Medicine and Health, Department of Laboratory Medicine, National Reference Laboratory for *Clostridioides difficile*, Clinical Microbiology, Örebro University, Örebro, Sweden, **3** 1928 Diagnostics, Göteborg, Sweden

* anders.e.werner@gmail.com

**Data Availability Statement:** All genome sequences in this study were deposited in the European Nucleotide Archive (ENA) under study

## Abstract

### Background

A zoonotic association has been suggested for several PCR ribotypes (RTs) of *Clostridioides difficile*. In central parts of Sweden, RT046 was found dominant in neonatal pigs at the same time as a RT046 hospital *C. difficile* infection (CDI) outbreak occurred in the southern parts of the country.

### Objective

To detect possible transmission of RT046 between pig farms and human CDI cases in Sweden and investigate the diversity of RT046 in the pig population using whole genome sequencing (WGS).

### Methods

WGS was performed on 47 *C. difficile* isolates from pigs (n = 22), the farm environment (n = 7) and human cases of CDI (n = 18). Two different core genome multilocus sequencing typing (cgMLST) schemes were used together with a single nucleotide polymorphisms (SNP) analysis and the results were related to time and location of isolation of the isolates.

### Results

The pig isolates were closely related (≤6 cgMLST alleles differing in both cgMLST schemes) and conserved over time and were clearly separated from isolates from the human hospital outbreak (≥76 and ≥90 cgMLST alleles differing in the two cgMLST schemes). However, two human isolates were closely related to the pig isolates, suggesting possible transmission. The SNP analysis was not more discriminate than cgMLST.

### Conclusion

No general pattern suggesting zoonotic transmission was apparent between pigs and humans, although contrasting results from two isolates still make transmission possible. Our

accession number PRJEB34857. Accession of individual isolates is presented in S1 Table.

**Funding:** K.J., P.M., T.N. and M.S. has received funding for this study from the Research Committee of Region Örebro County (grant OLL-674241). The funder had no role in study design, data collection and analysis, decision to publish, or preparation of manuscript. D.A. and F.D. employees of 1928 Diagnostics. 1928 Diagnostics provided support in the form of salaries for D.A. and F.D. and made the analysis available for free, but did not have any additional role in the study design, data collection and analysis, decision to publish, or preparation of the manuscript. The specific roles of these authors are articulated in the 'author contributions' section.

**Competing interests:** D.A. and F.D. are employees of 1928 Diagnostics. This does not alter our adherence to PLOS ONE policies on sharing data and materials. A.W., P.M., A.F., K.J., M.S., and T.N. declare no competing interests.

results support the need for high resolution WGS typing when investigating hospital and environmental transmission of *C. difficile*.

## Introduction

*Clostridioides difficile* (formerly *Clostridium difficile*) is a common cause of antibiotic-associated diarrhoea that causes significant mortality and morbidity as well as high costs for the healthcare system [1,2]. The spread of PCR ribotype (RT) 027 in healthcare settings at the beginning of the millennium put the spotlight on *C. difficile* infection (CDI) as a nosocomial disease [3]. In recent years, however, a rise in the incidence of community-associated CDI has been observed [4]. The most common methods for epidemiological investigation of *C. difficile*, such as PCR ribotyping and multilocus sequence typing (MLST), only offer a moderate resolution insufficient for outbreak investigation [5]. Use of core genome MLST (cgMLST) or single nucleotide polymorphisms (SNP) analysis to analyse data produced by whole genome sequencing (WGS) is more discriminate [6] and has revealed that transmission from symptomatic patients in the healthcare system only accounts for a small part of CDI cases. This suggests that asymptomatic carriage or environmental sources play an important role in the transmission of *C. difficile* [7].

*Clostridioides difficile* may also be carried by pigs and other livestock [8] and has been proposed as a cause of scouring among newborn piglets [9]. Potential transmission between livestock and humans has been described using WGS [10,11] and especially RT078 is considered to have a zoonotic potential [12]. Clusters of RT046 among humans have been reported in Poland and Chile [13,14], and this RT was one of the most frequently isolated among humans in Sweden in 2009–2013 [15]. In 2011 it was related to a hospital-based outbreak in southern Sweden [15]. During the same time, it was the only RT found among piglets at multiple breeding farms in central Sweden [16]. No zoonotic link has yet been established for RT046 and neither the clonal diversity, change over time within farms nor the relationship with human isolates is currently known.

The objective of this study was to detect possible transmission of RT046 between pig farms and human CDI cases in Sweden and investigate the RT046 diversity in the pig population using WGS. Multiple analysis strategies using two cgMLST schemes and one SNP analysis were performed.

## Materials and methods

### Sample selection

Forty-seven *C. difficile* isolates originating from pigs, the farm environment and human clinical cases were included (S1 Table). All human strains were retrieved prior to this study and PCR ribotyped as 046 in surveillance and routine programmes of the National Reference Laboratory for *C. difficile* either at the Public Health Agency of Sweden or at the Department of Laboratory Medicine, Clinical Microbiology, Örebro as described elsewhere [15]. The pig and environmental isolates were all PCR ribotyped as previously described [16]. Twenty isolates (ten collected in 2012 (P1–P10) and ten in 2017 (P11–P20)) originated from piglets and sows on a pig breeding farm in central Sweden (farm A). The sampling procedure has been described elsewhere [16]. Six environmental isolates from the surroundings of farm A (E1–E3, 2013 and E4–E6, 2017), two isolates collected from pigs at two other pig farms in the same county (P21, farm B, 2012 and P22, farm C, 2012) and one isolate isolated from a stream

approximately 1 km from farm C (E7, 2013) were included. Soil samples were collected in sterile containers for transportation; for water samples, 100 mL was collected. At the laboratory, approximately 2 g of soil was dissolved in 5 ml of sterile 0.85% NaCl solution, 2 mL of the soil mixture or 2 mL of water including sediment was transferred to the enrichment broth used for the pig isolates and then handled according to the protocol previously described [16].

Eighteen human CDI RT046 isolates (H1–H18), isolated between 2010 and 2017, were included. Ten were chosen from isolates sent for PCR ribotyping to the National Reference Laboratory for *C. difficile* because of suspicion of spread *C. difficile* in healthcare settings. Three of these (H2–H4) were from a previously described hospital-based outbreak [15]. The remaining eight human isolates were a subset of isolates from the yearly national survey on *C. difficile* run by the public health agency of Sweden [15] and chosen from diverse geographical sites.

## Ethical considerations

Clinical isolates were collected through routine programmes for typing of *C. difficile*. No patient information was collected. All other isolates had been previously collected and retrieved. Ethical approval was therefore not required.

## Culture conditions and DNA extraction

Isolates had been stored frozen (-80˚C) and were recovered on Fastidious Anaerobe Agar (Neogen®, Auchincruive, Scotland) with the addition of 5% horse blood. DNA from 24-hour cultures was manually extracted using the DNeasy UltraClean Microbial Kit (Qiagen, Hilden, Germany), with the following modified pre-step: one-third of a 10 μL loop of cultured bacteria was added to 300 μL PowerBead solution in a PowerBead tube (Qiagen). Thereafter, 50 μL SL solution (part of DNeasy UltraClean Microbial Kit) was added, vortexed briefly and incubated at 95˚C for 5 minutes. The tubes were vortexed for 20 minutes and then centrifuged at 10,000×g for 2 minutes and then processed according to the manufacturer's instructions and eluted in 50 μL.

## Library preparation and sequencing

Sequencing libraries were constructed using the Nextera XT DNA library prep kit (Illumina®, San Diego, CA, USA), with slight modifications to the protocol in order to optimize the average fragment length. The tagmentation was performed using lower input DNA, 0.75 ng, and the tagmentation time was increased to 7 minutes and 30 seconds. Amplification of the tagmented DNA was performed using index primers and the amplified products were automatically purified with the ACSIA NGS LibPrep Edition (PRIMADIAG, Romainville, France) using AMPure XP beads (Beckman Coulter, Brea, CA, USA). The normalization was performed using the ACSIA NGS LibPrep Edition. Pooling was performed manually based on the size of the fragments, determined by a TapeStation 4200 (Agilent, Santa Clara, CA, USA), and the DNA concentration, measured using a Qubit™ fluorometer (Thermo Fisher, Waltham, MA, USA).

Sequencing was performed using Illumina MiSeq™ (Illumina®) with MiSeq™ Reagent Kit v3 (600 cycles) (Illumina®), according to the manufacturer's instructions. More than 50-fold average coverage over the whole genome in combination with ≥97% of good cgMLST targets in Ridom™ SeqSphere+ version 6.0.2 (Ridom™ GmbH, Münster, Germany) was considered acceptable sequence quality for inclusion in all analyses (results for all isolates are listed in S1 Table). For analysis in Ridom™ SeqSphere+ the reads were *de novo* assembled through a pipeline using Velvet version 1.1.04 [17] using default settings and sequences were trimmed before

assembly until the average Phred quality score was 30 in a window of 20 bases. One MLST gene in isolate H4 was not assembled correctly by Velvet and was therefore assembled using SPAdes version 3.12.0 [18]; this assembly was used only for the MLST analysis in Ridom™ Seq-Sphere+. For the 1928D platform the raw sequences were trimmed from the 3' end until Phred quality score of 25 was reached. The 1928 platform's cgMLST method uses a custom developed allele calling algorithm based on an alignment free k-mer approach. For novel allele extraction, and database acceptance, local assembly is used to validate gene structure using SPAdes version 3.11.1 [18].

## Data analysis

The sequences were analysed by MLST and cgMLST using the software Ridom™ SeqSphere+, cgMLST scheme version 2.0 based on 2,147genes [19], and 1928D (1928 Diagnostics, Gothenburg, Sweden), cgMLST scheme based on 2,631 core genes (S2 Table) defined as genes that are present in 95% of 28 reference genomes (complete genomes available on National Center for Biotechnology Information (NCBI) by 11 July 2018). The seed genome for picking target genes belongs to strain 630 delta erm (NCBI RefSeq assembly accession number GCF_002080065.1). The 1928 cgMLST scheme was created with the purpose to be able to ana-lyse sequence types (STs) 1, 35, 3 and 37. ST 1 corresponds to hypervirulent RT027 [3], ST 35 corresponds to RT046 which historically has been common in Sweden [15] and STs 3 and 37 are common in certain demographic groups [20]. The scheme has also been observed to per-form well for STs 81, 41, 54 and 55. Further, the presence of genes in genomes was determined by a 90% similarity threshold. A limit of 97% (Ridom™ SeqSphere+) and 95% (1928D) of good cgMLST target genes was set for inclusion in the respective cgMLST analysis.

The SNP analysis was performed using the 1928D platform as a core genome alignment. 1928D's SNP analysis pipeline uses Burrows-Wheeler Aligner version 0.7.17-r1188 [21,22] for read alignment and FreeBayes version 1.3.2 [23] for variant calling. Isolate H10 was used as the reference genome as no public reference genome was available for RT046 [24]. Regions of genomes that aligned throughout all the samples and the reference genome were used to extract SNPs from. Variant calls were required to be homozygous, quality score ≥60 and sup-port by at least 10 high quality reads. Recombination regions were not identified or excluded. Toxin genes were identified using the 1928D platform.

## Results

All 47 isolates were categorized as ST35 by both Ridom™ SeqSphere+ and 1928D. Overall, the three analyses gave similar results regarding phylogenetic relationships (Figs 1–3). All isolates carried both toxin A and B genes (*tdc*A, *tdc*B). None of the isolates carried any of the binary toxin genes (*cdt*A, *cdt*B).

### Core genome multilocus sequencing typing analysis by Ridom™ SeqSphere+

The 47 isolates differed from 0 to 186 alleles in pairwise comparisons (S3 Table) and the major-ity of the human isolates were not closely related to the pig or the environmental isolates (Fig 1). Conversely, 28 out of the 29 isolates from pigs or the environment were closely related, with ≤6 cgMLST alleles differing, and were therefore considered to belong to the same cgMLST cluster (Fig 4) [19]. Within this cluster the isolates were distributed independent of sampling time or location. The remaining isolate E7, isolated in 2013, originated from a stream 1 km from farm C. It differed by 72 alleles from the closest pig or environmental isolate P21 (S3 Table), but showed only 2 allelic differences from one of the human isolates H13, isolated in 2016 (Fig 4). The isolates E7 and H13 originated from different geographical regions. Two

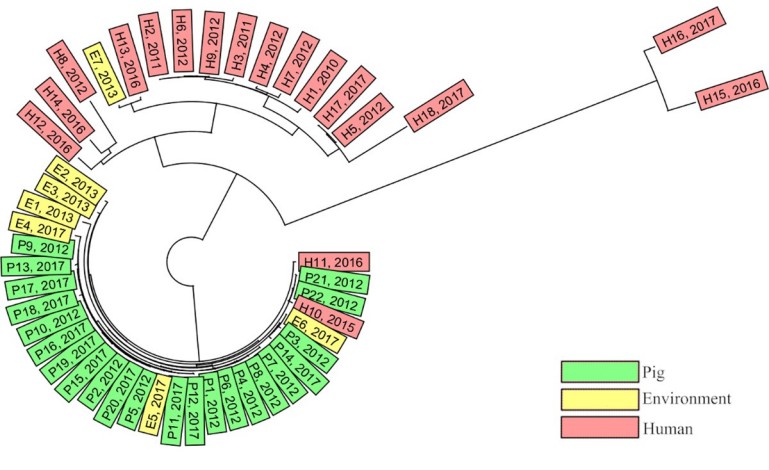

**Fig 1. Ridom™ SeqSphere+ cgMLST scheme tree.** Neighbour-joining tree of all isolates based on the Ridom™ SeqSphere+ core genome multilocus sequencing typing (cgMLST) scheme version 2 (2147 genes) and MLST scheme (7 genes), ignoring missing alleles, with year of isolation presented after the isolate's name and coloured according to the source of isolation.

human isolates H10 and H11, isolated in 2015 and 2016, differed by one and two alleles from the closest pig isolates P21 and P22 (Fig 4), these isolates were collected as a part of the yearly national survey and not in the same county as the pig farms are situated. All other human isolates differed by ≥65 (65–186) alleles from the 28 isolates in the pig and environmental cluster (S3 Table). Isolates H2–H4 from the previously described hospital outbreak [15] were all in the same cgMLST cluster. In addition, four clinical isolates were linked to this cluster (Fig 4). Three isolates were isolated during the time of the outbreak from a neighbouring county while the last isolate was collected approximately 200 km away the year before the outbreak was discovered. Among the other human isolates only one cluster, containing the two isolates H5 and H17, was observed. These isolates were collected 5 years apart in non-neighbouring counties. The greatest difference of 186 alleles was found between isolate H15 and isolates E1/E3/P13 (S3 Table).

## Core genome multilocus sequencing typing analysis by 1928D

The isolates differed from 0 to 238 alleles in pairwise comparisons (S4 Table) and the analysis showed very similar results to the Ridom™ SeqSphere+ analysis. All isolates belonging to the same cgMLST cluster in SeqSphere+ were grouped together in the analysis by 1928D and no new isolates were found to differ by ≤6 alleles (S4 Table). All the isolates in the large pig and environmental cluster, including the two intermingling human isolates, were grouped together and had ≤5 allelic differences. All seven isolates identified as related to the hospital outbreak were grouped by the 1928D analysis (≤2 allelic differences). Similarly, isolates E7/H13 and H5/H17 were also closely connected with 3 and 2 allelic differences, respectively (S4 Table). The greatest difference of 238 alleles was found between isolate H15 and isolates E1/E3 (S4 Table).

## Single nucleotide polymorphisms analysis

Good alignment quality was achieved for all isolates, with an average of 98.4% and a median of 98.6% of the genome aligned. This resulted in a core alignment of 3,666,694 core sites, corresponding to 90.3% of the reference genome H10, with 1,278 variant sites. The isolates had

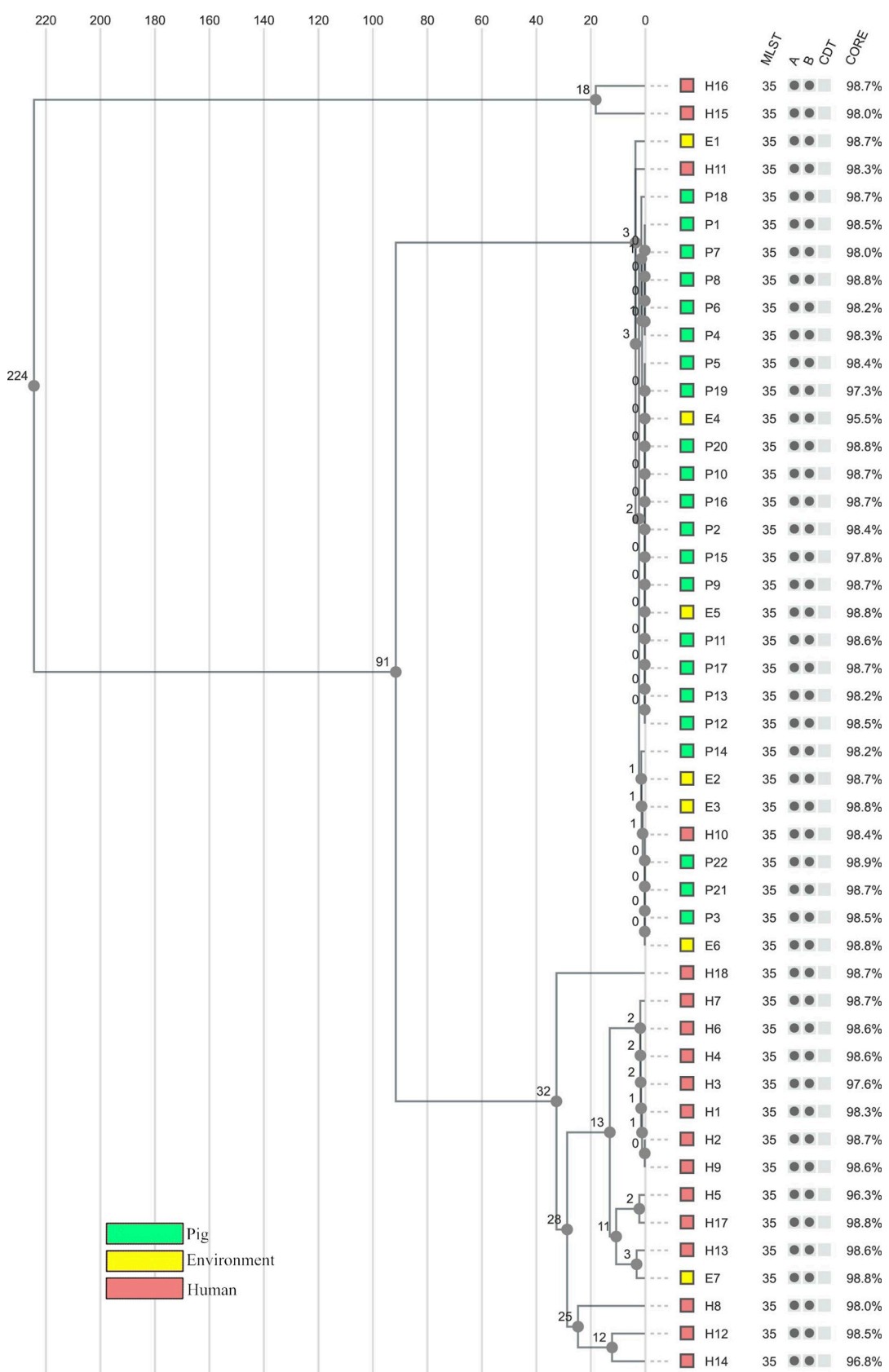

**Fig 2. 1928D cgMLST analysis tree.** Unweighted pair group method with arithmetic mean (UPGMA) tree including all isolates with distances based on the 1928D core genome multilocus sequencing typing (cgMLST) scheme (2,631 genes), ignoring missing alleles. Coloured according to the isolation source. Sequence type (MLST), the presence of toxin genes (A = *tdc*A, B = *tdc*B, CDT = binary toxin), and percentage of good cgMLST genes (CORE) are presented after each isolate.

between 0 and 899 SNP differences (S5 Table). A visual comparison of the tree structures from both cgMLST analyses (Figs 1 and 2) and the SNP analysis (Fig 3) showed overall similarity and the main clusters were identified. The 28 closely related pig and environmental isolates with the two intermingling human isolates displayed <10 SNP differences. The cluster containing the hospital outbreak isolates showed ≤6 SNP differences, with the three outbreak isolates H2–H4 having ≤5 SNP differences. The SNP analysis also confirmed the close relationship between isolate E7/H13 and H5/H17, with 3 and 7 SNP differences, respectively. The greatest difference of 899 SNP´s was found between isolate H15 and isolate P3 (S5 Table).

## Discussion

Our results confirm the previously reported low resolution of PCR ribotyping and MLST in *C. difficile* [5,10,25,26]. Core genome MLST separated the isolates into two distinct clusters, coherent with the epidemiological data available to us. One mainly consisted of isolates from the pig farms and the other which was associated with the human hospital outbreak. In both clusters, additional isolates with no known epidemiological association were included. The two different cgMLST schemes showed the same structure and very similar numbers of allelic differences within the clusters and therefore a similar performance within RT046 despite differences in the number of total alleles analysed and the algorithms used. The SNP analysis confirmed the close genetic similarity within the clusters and did not separate the isolates with known epidemiology from those with unknown epidemiology better than cgMLST. Applying the limit of >10 SNP differences for genetically distinct isolates, as suggested by Eyre et al [7], to our results corresponds to the limit of ≤6 allelic differences set for cgMLST complex types in Ridom™ SeqSphere+. Therefore, cgMLST seemed to offer the same resolution as SNP analysis. As the SNP analysis did not exclude recombination events, this indicates a low level of recombination within RT046. Similar performance of cgMLST and SNP analysis has been shown for other species [27–29] and as cgMLST offers a simpler workflow and the advantage of a fixed nomenclature within the same scheme. However, it has been shown that that the method used for assembly can significantly affect results of cgMLST and even cause the counterintuitive phenomenon of introducing false differences not supported by high quality SNP analysis [30]. We believe that cgMLST should be advocated for routine analysis of genetic relatedness in *C. difficile* but researchers and laboratories should be aware of the need to validate also the assembler software and that unexpected results might need to be confirmed with SNP analysis. It would be beneficial if a fixed cgMLST scheme was set for *C. difficile*, and other species, and used by different software providers to increase our understanding of worldwide epidemiology.

The close genetic similarity of the isolates from the pig farms, both between farms and over time, indicates a conserved population of RT046 in the sampled farms. The low rate of genetic mutations, indicated by the fact that isolates separated by 5 years were inseparable both in cgMLST and in SNP analysis, was striking and possibly due to dormant spore status. Importantly, the hospital outbreak [15] cluster was clearly separated from the pig and environmental cluster. The majority of the clinical isolates were more closely related to the hospital outbreak strains than to the pig and environmental strains (Fig 1), indicating that hospital transmission is an important route of acquisition of *C. difficile* RT046 in Sweden. However, two human

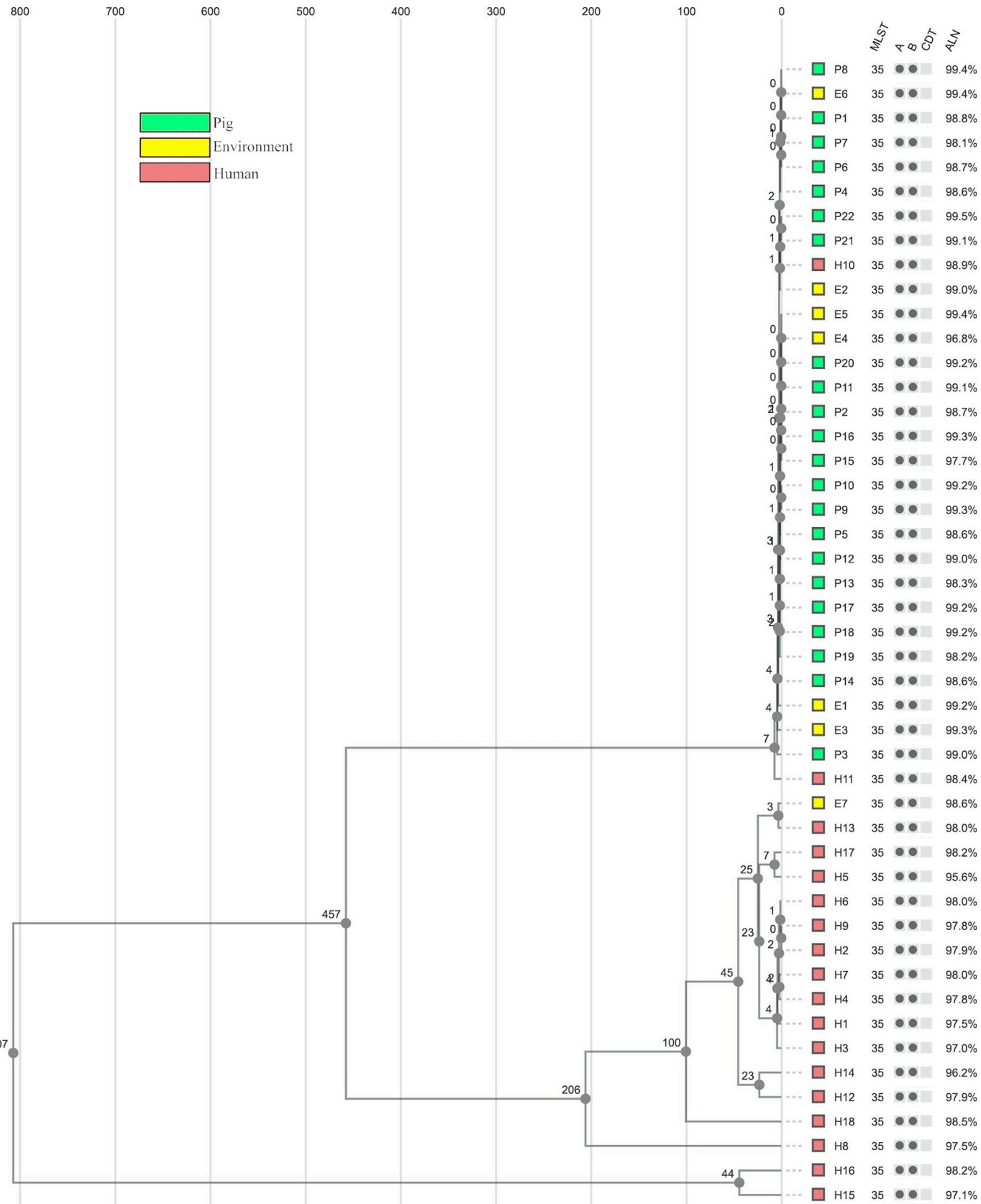

**Fig 3. SNP analysis tree.** Unweighted pair group method with arithmetic mean (UPGMA) tree including all isolates based on single nucleotide polymorphisms (SNP) differences in the 1928D analysis, coloured according to the isolation source. Sequence type (multilocus sequencing typing (MLST)), the presence of toxin genes (A = *tdc*A, B = *tdc*B, CDT = binary toxin), and the proportion of genome aligned (ALN) are presented after each isolate. The SNP analysis confirmed the clustering observed in both core genome (cg) MLST analyses.

isolates (H10 and H11) were intermingled in the pig and environmental cluster, indicating transmission of *C. difficile* between pigs and humans. This has also been observed for other RTs [10,12]. Knetsch et al suggest a bilateral transmission [12] of *C. difficile*, which in this study may be supported by the environmental isolate E7 which was far more similar to a human isolate than to the pig isolates. However, the high genetic stability over time in the population of *C. difficile* on the farms makes it difficult to draw any conclusions on the direction and possible time of transmission in our study.

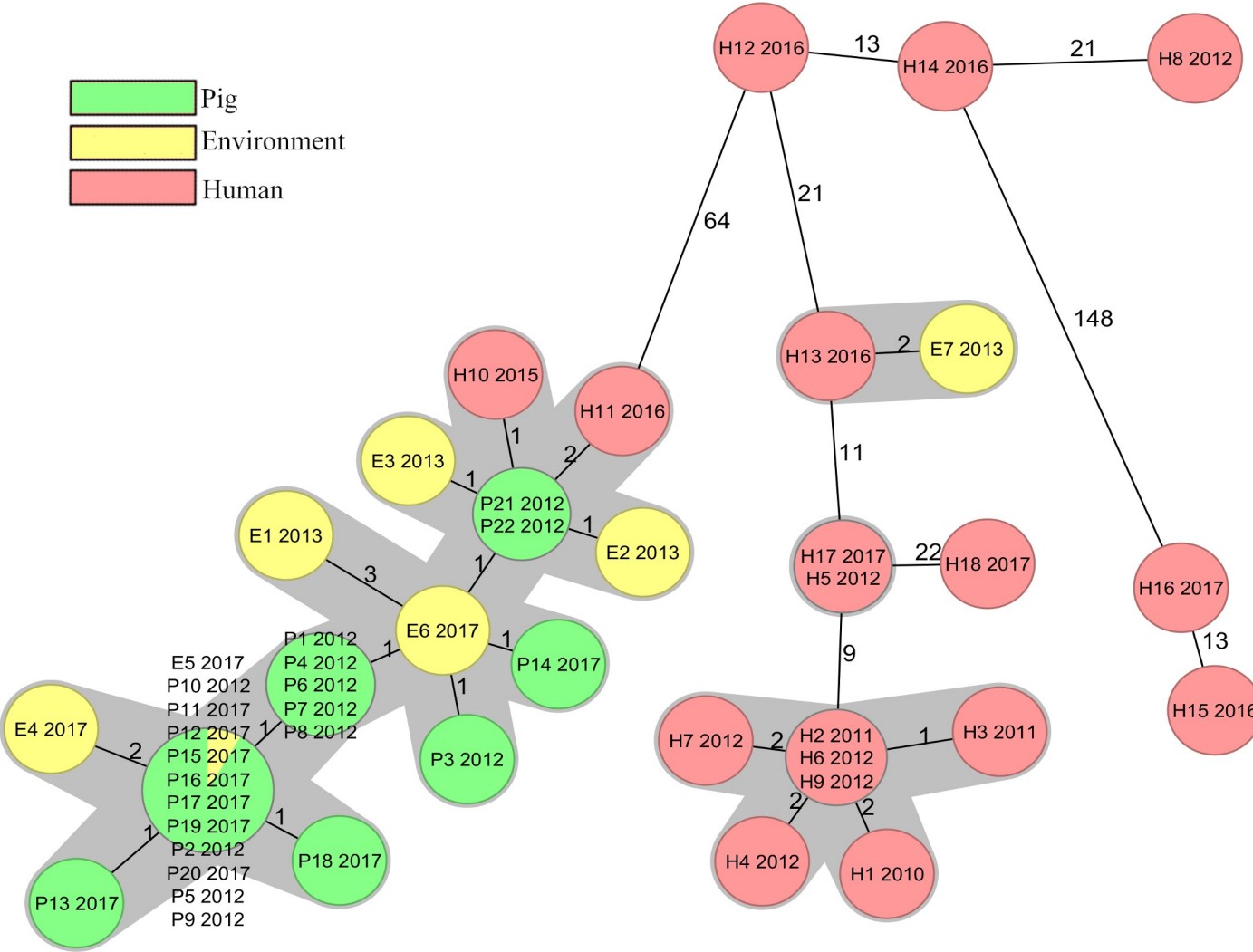

**Fig 4. Ridom™ SeqSphere+ cgMLST analysis minimum spanning tree.** Minimum spanning tree of all isolates based on the Ridom™ SeqSphere+ core genome multilocus sequencing typing (cgMLST) scheme version 2 (2147 genes) and MLST scheme (7 genes) with numbers of allelic differences shown on connecting lines (distances not to scale), ignoring missing alleles. Year of isolation is presented after the isolate's name, nodes are coloured according to the isolation source, and genetically closely related isolates (≤6 cgMLST alleles differing) are shaded grey.

The clonal spread and low mutation frequency [7,31] in *C. difficile*, combined with the ability to form spores, are factors that probably contribute to the similarity of isolates both over time and geographically. This similarity also includes isolates without known epidemiological links, as illustrated in our study by isolates H5/H17 and E7/H13. Unexplained close relationships like this have also been described by others [7,10,12]. Of course, unknown common sources and ways of transmission must be considered. The risk of unknown epidemiological connections is also one of the limitations of this study. Only basic epidemiological data was available, such as year of isolation, source of the isolate and whether some of the isolates were isolated during ongoing transmission. Additionally, a selection of the routinely collected isolates was performed to achieve a high diversity in the collection of human isolates, which may have hampered the possibility to reveal local spread. Another limitation of our study is that *C. difficile* has a relatively small proportion of core genome compared with other species [32] and that all the analyses focus on differences in the core genome (also for the SNP analysis, although with a larger core). Muñoz et al [33] found incongruence in the phylogenetic tree topology when comparing core genome with accessory genome analysis. Since the core genome is so stable over time [32], changes in the accessory genome could in theory be more discriminating. Long read sequencing, which is now becoming more accessible, could in the future help to further increase the discriminatory power of WGS in *C. difficile* by making it easier to track mobile elements in the genome [24].

To conclude, this study analysed RT046 *C. difficile* isolated from different sources in Sweden from 2010 to 2017. No close connection between isolates from pig farms and a large hospital outbreak was found, but two human isolates (not related to the outbreak) clustered together with pig isolates, suggesting zoonotic transmission. Two different cgMLST schemes showed similar results and an SNP analysis yielded similar genetic resolution. Our study confirms the need for high-resolution typing (i.e. WGS) to map transmission of *C. difficile* both in hospital and in the surrounding environment.

## Supporting information

**S1 Table. Summary of all isolates.** All isolates presented with location of isolation, year of isolation, European Nucleotide Archive accession, average coverage, and percentage of good cgMLST targets.
(XLSX)

**S2 Table. 1928 Core genes.** List of core genes of the 1928 diagnostic platform core genome multilocus sequence typing scheme, gene names are the annotated genes from the strain 630 delta erm seed genome(NCBI RefSeq assembly accession number GCF_002080065.1) and all annotated genes and their exact position can be found on the chromosomal unit of the genome (NCBI RefSeq assembly accession number NZ_CP016318.1).
(XLSX)

**S3 Table. Ridom SeqSphere+ cgMLST distance matrix.** Distance matrix showing pairwise comparison of allelic differences between all 47 isolates based on the 2,147 core genes in the Ridom SeqSphere+ core genome multilocus sequence typing (cgMLST) scheme and 7 MLST genes.
(XLSX)

**S4 Table. 1928 Diagnostics cgMLST distance matrix.** Distance matrix showing pairwise comparison of allelic differences between all 47 isolates based on the 2,631 core genes in the 1928 Diagnostics core genome multilocus sequence typing (cgMLST) scheme.
(XLSX)

**S5 Table. SNP matrix.** Single nucleotide polymorphism matrix over all isolates based on the 1928 diagnostics analysis.
(XLSX)

## Acknowledgments

We would like to thank the Public Health Agency of Sweden and especially Kristina Rizzardi and Thomas Åkerlund for access to isolates, as well as all laboratories in Sweden that contribute to the national surveillance programme for *C. difficile*. Our thanks also go to Andreas Matussek and colleagues at the Department of Clinical Microbiology of the County Hospital Ryhov, Jönköping, for access to isolates, and Theresa Ennefors for technical assistance.

## Author Contributions

**Conceptualization:** Anders Werner, Paula Mölling, Karin Johansson, Martin Sundqvist, Torbjörn Norén.

**Formal analysis:** Anders Werner, Paula Mölling, Anna Fagerström, Fredrik Dyrkell, Dimitrios Arnellos.

**Funding acquisition:** Paula Mölling, Karin Johansson, Martin Sundqvist, Torbjörn Norén.

**Methodology:** Paula Mölling, Anna Fagerström.

**Writing – original draft:** Anders Werner, Paula Mölling, Martin Sundqvist, Torbjörn Norén.

**Writing – review & editing:** Paula Mölling, Anna Fagerström, Fredrik Dyrkell, Dimitrios Arnellos, Karin Johansson, Martin Sundqvist, Torbjörn Norén.

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
