## [Decision Letter · Decision Letter 0]

8 Jul 2020

PONE-D-20-18714

Whole genome sequencing of Clostridioides difficile PCR ribotype 046 suggests transmission between pigs and humans

PLOS ONE

Dear Dr. Werner,

Thank you for submitting your manuscript to PLOS ONE. After careful consideration, we feel that it has merit but does not fully meet PLOS ONE’s publication criteria as it currently stands. Therefore, we invite you to submit a revised version of the manuscript that addresses the points raised during the review process.

Your manuscript has been reviewed by tow experts in your field. based on their comments, a major revision is needed before a decision can be made.

We look forward to receiving your revised manuscript.

Kind regards,

Yung-Fu Chang

Academic Editor

PLOS ONE

Journal Requirements:

2. We note that you are reporting an analysis of a microarray, next-generation sequencing, or deep sequencing data set. PLOS requires that authors comply with field-specific standards for preparation, recording, and deposition of data in repositories appropriate to their field. Please upload these data to a stable, public repository (such as ArrayExpress, Gene Expression Omnibus (GEO), DNA Data Bank of Japan (DDBJ), NCBI GenBank, NCBI Sequence Read Archive, or EMBL Nucleotide Sequence Database (ENA)). In your revised cover letter, please provide the relevant accession numbers that may be used to access these data. For a full list of recommended repositories, see http://journals.plos.org/plosone/s/data-availability#loc-omics or http://journals.plos.org/plosone/s/data-availability#loc-sequencing.

3. Thank you for stating the following in the Competing Interests:

"D.A. and F.D. are employees of 1928 Diagnostics. A.W., P.M., A.F., K.J., M.S., and T.N. declare no conflict of interest."

We note that one or more of the authors are employed by a commercial company: 1928 Diagnostics.

3.1. Please provide an amended Funding Statement declaring this commercial affiliation, as well as a statement regarding the Role of Funders in your study. If the funding organization did not play a role in the study design, data collection and analysis, decision to publish, or preparation of the manuscript and only provided financial support in the form of authors' salaries and/or research materials, please review your statements relating to the author contributions, and ensure you have specifically and accurately indicated the role(s) that these authors had in your study. You can update author roles in the Author Contributions section of the online submission form.

3.2. Please also provide an updated Competing Interests Statement declaring this commercial affiliation along with any other relevant declarations relating to employment, consultancy, patents, products in development, or marketed products, etc. 

Reviewers' comments:

Reviewer's Responses to Questions

**Comments to the Author**

1. Is the manuscript technically sound, and do the data support the conclusions?

Reviewer #1: Partly

Reviewer #2: Yes

2. Has the statistical analysis been performed appropriately and rigorously? 

Reviewer #1: I Don't Know

Reviewer #2: N/A

3. Have the authors made all data underlying the findings in their manuscript fully available?

Reviewer #1: No

Reviewer #2: Yes

4. Is the manuscript presented in an intelligible fashion and written in standard English?

Reviewer #1: Yes

Reviewer #2: Yes

5. Review Comments to the Author

Reviewer #1: PONE-D-20-18714

This paper compares genome sequences from C. difficile ribotype 046 isolates collected from pigs and humans by using SNP typing and two different schemes for cgMLST, one of which is novel. While some of the results may be interesting, the level of methodological detail given is insufficient to fully assess the conclusions presented.

Line 124: "For analysis in Ridom Seqsphere .... the reads were de novo assembled ... using Velvet ..." Please provide the version and parameter settings of the assembler software.

Line 127: How were the reads assembled for analysis on the 1928D platform?

Line 137: A "cgMLST scheme based on 2,631 core genes" apparently was used for analysis with the 1928D software. This cgMLST scheme appears to be novel. However, the structure of this cgMLST scheme is not reported anywhere in the manuscript, nor is any literature reference provided. Clearly, without more detailed information on this method, the results based on this new cgMLST which are presented here are of little use to the reader. At the least, information on the genes included in the scheme and the exact positions of gene sequences in relation to a publically available reference genome sequence must be provided.

It is also not clear if the novel cgMLST scheme was evaluated in any way, e.g. by applying it to a reference set of genomes or strains.

Line 143: "The SNP analysis was performed using the 1928D platform" -- Unclear, which strategy was used for SNP analysis. Please provide transparent detail on the algorithm/software and parameters used.

Line 146: "Variants were called at 10x minimum coverage" -- Such low coverage is usually not considered sufficient for calling SNPs. Why was a different minimum coverage set in comparison to SeqSphere cgMLST analysis (50x, see Line 129)?

Line 147: Genomic regions affected by recombination may confound SNP-based phylogenetic analyses and therefore commonly are excluded. Why did the authors choose not to do so?

Line 155, Figure 1: The structure within the main clade, including all the pig isolates, cannot be discerned.

Line 159, Figure 2: It is difficult to compare the trees in Figure 1 (SeqSphere) and Figure 2 (1982D), because the former was drawn in a circular format and the other was drawn rectangular. Please provide both trees in the same format.

Line 233: "very similar numbers of allelic differences within the clusters" -- Are the numbers of allelic differences shown anywhere in the manuscript? This would be interesting.

Line 235: "differences in the algorithms used" [by SeqSphere and 1928D] -- The algorithm used by 1928D needs to be provided.

Line 242: "Similar performance of cgMLST and SNP analysis" -- Is this really true? Eyre et al. (J. Clin. Microbiol. 58: 01037-19) recently showed that cgMLST was inferior to SNP analysis for identifying closely related C. difficile isolates.

Line 257: "two human isolates were intermingled in the pig and environmental cluster" -- Where and when had these human specific isolates been collected? Had they been part of the national survey, with no connection to the area around the pig farms? Please provide this information in the manuscript.

The sequence data is not available at ENA under the accession number indicated.

Reviewer #2: In the present manuscript authors describe results of WGS comparison of 47 strains obtained from humans, pigs and the farm environment, using two different approaches, cgMLST, with two different schemes, and SNV analysis.

It is an interesting paper, but I do have some comments.

Add more detailed description of cgMLST on 1928D platform. How were the sequences assembled, which assembly software was used.

There in an updated Ridom SeqSphere cgMLST scheme that was released recently, please re-run the analysis with the updated scheme.

Lines 126 and 127: when were sequences trimmed, before or after assembly?

Add more info on what parameters were used for assessing the SNV. How were the genomes assembled, were there any quality trimming applied before?

Figure 1 and 2. Add number of different alleles between genomes. Also, mark the CC (at least for SeqSphere I know that ST that differ in less than 6 alleles can be shaded).

To maybe improve the resolution you could also include part (core) of the accessory genome for the cgMLST analysis – this can be done using SeqSphere.

Line 53-54: change the sentence…in this paper Eyre et al. did not unequivocally show that transmission outside the healthcare system in an important way for acquisition of CD. They suggested that there is a large, genetically diverse reservoir outside the hospital setting.

Add info on ENA accession number to Materials/Method section.

Line 174 and 204: Change …by 0 and 204… to …from 0 to 204….

6. PLOS authors have the option to publish the peer review history of their article (what does this mean?). If published, this will include your full peer review and any attached files.

Reviewer #1: No

Reviewer #2: No

---

## [Author Response · Author response to Decision Letter 0]

18 Oct 2020

Response to reviewer and academic editor.

Thank you for reviewing our manuscript and for the constructive comments.

We now hope we have corrected the style requirements we missed the first time around.

Regarding the availability of data, all our data have always been deposited in the EMBL Nucleotide Sequence Database (ENA) but held private until we felt that we were close to being published and we have now made all sequence data public available. The sequences can as stated in the manuscript be found under project accession number PRJEB34857 (individual sequence accession numbers available in S1 Table).

Thank you for clarifying how you want the competing interest and funding statement to be expressed. We would like to adjust them to the following:

Funding statement:

K.J., P.M., T.N. and M.S. has received funding for this study from the Research

Committee of Region Örebro County (grant OLL-674241). The funder had no role in study design, data collection and analysis, decision to publish, or preparation of manuscript.

D.A. and F.D. employees of 1928 Diagnostics. 1928D diagnostics provided support in the form of salaries for D.A. and F.D. and made the analysis available for free, but did not have any additional role in the study design, data collection and analysis, decision to publish, or preparation of the manuscript. The specific roles of these authors are articulated in the ‘author contributions’ section.

Competing interest statement.

D.A. and F.D. are employees of 1928 Diagnostics. This does not alter our adherence to PLOS ONE policies on sharing data and materials. A.W., P.M., A.F., K.J., M.S., and T.N. declare no competing interests.

To address that PLOS ONE do not allow data not shown as reference we have added two more distances matrixes as supplementary files (S3 Table and S4 Table) covering all the data formerly referred to as data not shown. 

Below we have responded to each question raised by the reviewers, for clarity the reviewer’s questions are marked in bold lettering. 

Reviewer #1:

This paper compares genome sequences from C. difficile ribotype 046 isolates collected from pigs and humans by using SNP typing and two different schemes for cgMLST, one of which is novel. While some of the results may be interesting, the level of methodological detail given is insufficient to fully assess the conclusions presented.

Additional methodological information has been inserted into text as well as supplementary items for better understanding. Care has also been taken not to make the methodological issues comparing the different methods to overcast the findings of how the isolates relate to each other.

Line 124: "For analysis in Ridom Seqsphere .... the reads were de novo assembled ... using Velvet ..." Please provide the version and parameter settings of the assembler software.

We have changes the sentence to: “For analysis in Ridom™ SeqSphere+ the reads were de novo assembled through a pipeline using Velvet version 1.1.04 [17] using default settings and sequences were trimmed before assembly until the average Phred quality score was 30 in a window of 20 bases.”

Line 127: How were the reads assembled for analysis on the 1928D platform?

We have added the following text to clarify the assembly in the 1928D platform “The 1928 platform’s cgMLST method uses a custom developed allele calling algorithm based on an alignment free k-mer approach. For novel allele extraction, and database acceptance, local assembly is used to validate gene structure using SPAdes version 3.11.1 [18].”

Line 137: A "cgMLST scheme based on 2,631 core genes" apparently was used for analysis with the 1928D software. This cgMLST scheme appears to be novel. However, the structure of this cgMLST scheme is not reported anywhere in the manuscript, nor is any literature reference provided. Clearly, without more detailed information on this method, the results based on this new cgMLST which are presented here are of little use to the reader. At the least, information on the genes included in the scheme and the exact positions of gene sequences in relation to a publically available reference genome sequence must be provided. It is also not clear if the novel cgMLST scheme was evaluated in any way, e.g. by applying it to a reference set of genomes or strains.

We have added a supplementary table (S2 Table) that contains a list of all core genes in the 1928D cgMLST scheme with gene names derived from the strain 630 delta erm seed genome. And added the text: “The seed genome for picking target genes belongs to strain 630 delta erm (NCBI RefSeq assembly accession number GCF_002080065.1). The 1928 cgMLST scheme was created with the purpose to be able to analyse sequence types (STs) 1, 35, 3 and 37. ST 1 corresponds to hypervirulent RT027 [3], ST 35 corresponds to RT046 which historically has been common in Sweden [15] and STs 3 and 37 are common in certain demographic groups [20]. The scheme has also been observed to perform well for STs 81, 41, 54 and 55.” This text also contains a new reference 

Line 143: "The SNP analysis was performed using the 1928D platform" -- Unclear, which strategy was used for SNP analysis. Please provide transparent detail on the algorithm/software and parameters used.

We have added the text “1928D’s SNP analysis pipeline uses Burrows-Wheeler Aligner version 0.7.17-r1188 [21, 22] for read alignment and FreeBayes version 1.3.2 [23] for variant calling.” and references for the software used. 

Line 146: "Variants were called at 10x minimum coverage" -- Such low coverage is usually not considered sufficient for calling SNPs. Why was a different minimum coverage set in comparison to SeqSphere cgMLST analysis (50x, see Line 129)?

We have changed some of the wording in the sentence in line 146 to clarify, but we maintain that it is sufficient for variant calling. In the reference referred to by reviewer #1 in a comment on line 242 they used a minimum of 5 reads to call a SNP, and Bush et al. (Genomic diversity affects the accuracy of bacterial single-nucleotide polymorphism–calling pipelines, GigaScience, Volume 9, Issue 2, February 2020) also used a minimum of 5 reads, a lover threshold than ours. The minimum 50x average coverage referred to in line 129 is an average coverage for the whole genome and is used as a measurement on over all sequence quality and the minimum 10x coverage is the coverage for a specific base to ensure that we only have high quality SNP included in our analysis. To clarify this we also have changed the wording in line 129. 

Line 147: Genomic regions affected by recombination may confound SNP-based phylogenetic analyses and therefore commonly are excluded. Why did the authors choose not to do so?

Manual analysis of recombination filtering results using Gubbins (v2.3.4) resulted in minimal variant differences that neither affected topology of the tree nor the relations between samples, the evaluation used the analysis results coming out directly from the 1928 platform.

Line 155, Figure 1: The structure within the main clade, including all the pig isolates, cannot be discerned.

Since the pig isolates are so closely related it is hard to visualise them in a three structure with all isolates included, it is for this reason that Fig 4 is a minimum spanning tree. We tried many different options and found that minimum spanning tree was the best way to present the data. To hopefully clarify the relationship between the isolate we also made figure 4 to include all isolates when we remade the figure in version 2 of the Ridom SeqSphere cgMLST scheme. 

Line 159, Figure 2: It is difficult to compare the trees in Figure 1 (SeqSphere) and Figure 2 (1982D), because the former was drawn in a circular format and the other was drawn rectangular. Please provide both trees in the same format.

We have tried both configurations of the tree for figure 1 and we feel that for the Ridom SeqSphere analysis the circular is easiest to read and shows the clusters in relation to each other in a good way. Unfortunately, it is not possible to make figure 2 circular. We feel that making each figure as easy to read as possible is preferable even with the small disadvantage of having the trees configured in different ways. 

Line 233: "very similar numbers of allelic differences within the clusters" -- Are the numbers of allelic differences shown anywhere in the manuscript? This would be interesting. 

All pairwise comparisons for all three analyses are now available in S3 Table, S4 Table and S5 Table

Line 235: "differences in the algorithms used" [by SeqSphere and 1928D] -- The algorithm used by 1928D needs to be provided.

Please see our answers to line 127 and 143

Line 242: "Similar performance of cgMLST and SNP analysis" -- Is this really true? Eyre et al. (J. Clin. Microbiol. 58: 01037-19) recently showed that cgMLST was inferior to SNP analysis for identifying closely related C. difficile isolates.

The sentence in line 242 refers to other species than C. difficile but we have read the article referred to and found it a good reference to include in our article in the discussion section. It highlights some of the difficulties in analysing whole genome sequencing data and we think it strengthens our decision to include more than one analysis scheme in the article including a SNP analysis. 

Line 257: "two human isolates were intermingled in the pig and environmental cluster" -- Where and when had these human specific isolates been collected? Had they been part of the national survey, with no connection to the area around the pig farms? Please provide this information in the manuscript.

The isolates are discussed in the results section to make the connection clearer for the reader we have named the isolates in the discussion section. The time of isolation was already named in the results section but we have also added the following text “these isolates were collected as a part of the yearly national survey and not in the same county as the pig farms are situated”.

The sequence data is not available at ENA under the accession number indicated.

This has been corrected as described above as a response to the editor’s questions. 

Reviewer #2:

Add more detailed description of cgMLST on 1928D platform. How were the sequences assembled, which assembly software was used.

Please see our answer to reviewer #1 Line 137. 

There in an updated Ridom SeqSphere cgMLST scheme that was released recently, please re-run the analysis with the updated scheme.

We have now reanalysed the Ridom Seqsphere analysis with version 2 of the cgMLST scheme and updated all number of allelic differences in the manuscript and updated figures 1 and 4. 

Lines 126 and 127: when were sequences trimmed, before or after assembly?

The sequences were trimmed before assembly, this information has been added to the manuscript. 

Add more info on what parameters were used for assessing the SNV. How were the genomes assembled, were there any quality trimming applied before?

Please see our answer to reviewer #1 Line 143

Figure 1 and 2. Add number of different alleles between genomes. Also, mark the CC (at least for SeqSphere I know that ST that differ in less than 6 alleles can be shaded).

To configure the figures so they are easy to read in the article is challenging and we have tried multiple configurations in Ridom SeqSphere and found that adding the requested information makes them harder to interpret. All allelic differences can be found in the now added tables S3 and S4. In Fig 4 we display the number of allelic differences in the minimum spanning trees for the Ridom SeqSphere+ analysis and the CC is shaded, we believe that from this figure the most important information can be extracted. 

To maybe improve the resolution you could also include part (core) of the accessory genome for the cgMLST analysis – this can be done using SeqSphere.

We appreciate the suggestion and have looked in to it but we believe this to be outside the scope of this article since it would make the focus even more on compering different typing methods and our primary focus is intended to be the relationship between the isolates. We have tried analysing the isolates with the part accessory genome scheme in Ridom SeqSphere and it did not change any of our findings. 

Line 53-54: change the sentence…in this paper Eyre et al. did not unequivocally show that transmission outside the healthcare system in an important way for acquisition of CD. They suggested that there is a large, genetically diverse reservoir outside the hospital setting.

We have changed the sentence to” and has revealed that transmission from symptomatic patients in the healthcare system only accounts for a small part of CDI cases, this suggests that asymptomatic carriage or environmental sources play an important role in the transmission of C. difficile CDI [7]”

Add info on ENA accession number to Materials/Method section.

As we have interpreted the submission guidelines the data availability section will be typeset with the article and to repeat the accession number in the method section feels unnecessary so at the moment we have made no changes. But if it is required to do so we will gladly add a paragraph with the accession number. 

Line 174 and 204: Change …by 0 and 204… to …from 0 to 204….

We have changed this according to the reviewer’s suggestion.

We would like to thank the reviewers and editor for their input, we feel they have greatly improved our manuscript.

Yours sincerely 

Anders Werner

---

## [Decision Letter · Decision Letter 1]

6 Nov 2020

PONE-D-20-18714R1

Whole genome sequencing of Clostridioides difficile PCR ribotype 046 suggests transmission between pigs and humans

PLOS ONE

Dear Dr. Werner,

Thank you for submitting your manuscript to PLOS ONE. After careful consideration, we feel that it has merit but does not fully meet PLOS ONE’s publication criteria as it currently stands. Therefore, we invite you to submit a revised version of the manuscript that addresses the points raised during the review process.

Your revised manuscript has been re-reviewed by the original reviewers and a major revision is still suggested.

We look forward to receiving your revised manuscript.

Kind regards,

Yung-Fu Chang

Academic Editor

PLOS ONE

Reviewers' comments:

Reviewer's Responses to Questions

**Comments to the Author**

1. If the authors have adequately addressed your comments raised in a previous round of review and you feel that this manuscript is now acceptable for publication, you may indicate that here to bypass the “Comments to the Author” section, enter your conflict of interest statement in the “Confidential to Editor” section, and submit your "Accept" recommendation.

Reviewer #1: (No Response)

Reviewer #2: All comments have been addressed

2. Is the manuscript technically sound, and do the data support the conclusions?

Reviewer #1: Partly

Reviewer #2: Yes

3. Has the statistical analysis been performed appropriately and rigorously? 

Reviewer #1: I Don't Know

Reviewer #2: N/A

4. Have the authors made all data underlying the findings in their manuscript fully available?

Reviewer #1: No

Reviewer #2: Yes

5. Is the manuscript presented in an intelligible fashion and written in standard English?

Reviewer #1: Yes

Reviewer #2: Yes

6. Review Comments to the Author

Reviewer #1: My most important point of critique of this manuscript is that it is based on a novel algorithm for cgMLST analysis that is insufficiently explained. While the authors have now added a supplementary table (S2_Table.xlsx) in response to my previous request, this table merely provides a list of gene names, which is useless to any reader without information on the precise positions of the sequence stretches that got evaluated. I had requested the same information in my previous review (see below). It seems that the authors wish to keep this information proprietary for any reason, but in that case the method cannot be reproduced by any other researchers and the results cannot be usefully compared to those from previously published cgMLST methods for C. difficile.

Previous comment:

Line 137: A "cgMLST scheme based on 2,631 core genes" apparently was used for

analysis with the 1928D software. This cgMLST scheme appears to be novel.

However, the structure of this cgMLST scheme is not reported anywhere in the

manuscript, nor is any literature reference provided. Clearly, without more detailed

information on this method, the results based on this new cgMLST which are presented

here are of little use to the reader. At the least, information on the genes included in the

Powered by Editorial Manager® and ProduXion Manager® from Aries Systems Corporation

scheme and the exact positions of gene sequences in relation to a publically available

reference genome sequence must be provided. It is also not clear if the novel cgMLST

scheme was evaluated in any way, e.g. by applying it to a reference set of genomes or

strains.

Previous response:

We have added a supplementary table (S2 Table) that contains a list of all core genes

in the 1928D cgMLST scheme with gene names derived from the strain 630 delta erm

seed genome.

Reviewer #2: (No Response)

7. PLOS authors have the option to publish the peer review history of their article (what does this mean?). If published, this will include your full peer review and any attached files.

Reviewer #1: No

Reviewer #2: No

---

## [Author Response · Author response to Decision Letter 1]

25 Nov 2020

Thank you for reviewing our manuscript again. 

If we have understood the critique correctly you want us to specify the exact position in the reference genome of each gene, thereby providing the reader with the sequences of all genes used in the cgMLST scheme. The gene names we have enclosed in S2 Table are the annotated gene names from the reference genome, so the exact position for each gene is publicly available in NCBI. It has never been our intention to keep this information proprietary. Contrary, the reasoning behind providing the annotated gene names is to take advantage of a publicly available nomenclature both to reduce the amount of information needed to be recited in the article and to make everything as transparent as possible. 

To make this clearer to the reader vi have changed the supporting information legend of S2 Table to “S2 Table. 1928 Core genes. List of core genes of the 1928 diagnostic platform core genome multilocus sequence typing scheme, gene names are the annotated genes from the strain 630 delta erm seed genome(NCBI RefSeq assembly accession number GCF_002080065.1) All annotated genes and their exact positions can be found on the chromosomal unit of the genome (NCBI RefSeq assembly accession number NZ_CP016318.1).” We added the reference number for the chromosomal unit of the genome since it has to be selected to be able to browse through the genes. 

If we have misunderstood the critique in any way or there are any further unclarities we will gladly try our best to explain our work.

Kind regards 

Anders Werner

---

## [Decision Letter · Decision Letter 2]

7 Dec 2020

Whole genome sequencing of Clostridioides difficile PCR ribotype 046 suggests transmission between pigs and humans

PONE-D-20-18714R2

Dear Dr. Werner,

We’re pleased to inform you that your manuscript has been judged scientifically suitable for publication and will be formally accepted for publication once it meets all outstanding technical requirements.

Kind regards,

Yung-Fu Chang

Academic Editor

PLOS ONE

Additional Editor Comments (optional):

Reviewers' comments:

Reviewer's Responses to Questions

**Comments to the Author**

1. If the authors have adequately addressed your comments raised in a previous round of review and you feel that this manuscript is now acceptable for publication, you may indicate that here to bypass the “Comments to the Author” section, enter your conflict of interest statement in the “Confidential to Editor” section, and submit your "Accept" recommendation.

Reviewer #1: All comments have been addressed

2. Is the manuscript technically sound, and do the data support the conclusions?

Reviewer #1: Yes

3. Has the statistical analysis been performed appropriately and rigorously? 

Reviewer #1: Yes

4. Have the authors made all data underlying the findings in their manuscript fully available?

Reviewer #1: Yes

5. Is the manuscript presented in an intelligible fashion and written in standard English?

Reviewer #1: Yes

6. Review Comments to the Author

Reviewer #1: (No Response)

7. PLOS authors have the option to publish the peer review history of their article (what does this mean?). If published, this will include your full peer review and any attached files.

Reviewer #1: No

---

## [Editor Report · Acceptance letter]

11 Dec 2020

PONE-D-20-18714R2 

Whole genome sequencing of *Clostridioides difficile* PCR ribotype 046 suggests transmission between pigs and humans 

Dear Dr. Werner:

I'm pleased to inform you that your manuscript has been deemed suitable for publication in PLOS ONE. Congratulations! Your manuscript is now with our production department. 

Kind regards, 

on behalf of

Dr. Yung-Fu Chang 

Academic Editor

PLOS ONE